

# Exact entanglement in the driven quantum symmetric simple exclusion process

### Denis Bernard[*] and Ludwig Hruza[†]

Laboratoire de Physique de l'École Normale Supérieure, CNRS, ENS & PSL University,
Sorbonne Université, Université Paris Cité, 75005 Paris, France

[*] denis.bernard@ens.fr , [†] ludwig.hruza@ens.fr

## Abstract

Entanglement properties of driven quantum systems can potentially differ from the equilibrium situation due to long range coherences. We confirm this observation by studying a suitable toy model for mesoscopic transport: the open quantum symmetric simple exclusion process (QSSEP). We derive exact formulae for its mutual information between different subsystems in the steady state and show that it satisfies a volume law. Surprisingly, the QSSEP entanglement properties only depend on data related to its transport properties and we suspect that such a relation might hold for more general mesoscopic systems. Exploiting the free probability structure of QSSEP, we obtain these results by developing a new method to determine the eigenvalue spectrum of sub-blocks of random matrices from their so-called local free cumulants – a mathematical result on its own with potential applications in the theory of random matrices. As an illustration of this method, we show how to compute expectation values of observables in systems satisfying the Eigenstate Thermalization Hypothesis (ETH) from the local free cumulants.



# 1  Introduction

In classical information theory, the mutual information between two correlated random variables $X$ and $Y$ with joint distribution $p_{XY}$ quantifies the information we gain about $X$ when learning $Y$,

$$I(X:Y) = H(p_X) + H(p_Y) - H(p_{XY}),$$

with $H(p) = -\int p(x)\log p(x)\,dx$ the Shannon entropy of the distribution $p$. A useful communication channel is therefore one in which the mutual information between input and output is high. In this sense the mutual information characterizes the physical properties of the channel itself.

From a condensed matter perspective, mutual information and entropy are useful tools to characterize the physical properties of many-body quantum systems, where the role of classical correlations is played by entanglement: For example, in systems with local interactions the entanglement entropy $S(\rho_I) = -\mathrm{Tr}(\rho_I \log \rho_I)$ of a subregion $I$ in the ground state usually scales with the boundary area of the subregion [1] – and not with its volume, as one would expect for typical energy eigenstates whose entanglement entropy is proportional to their thermodynamic entropy [2] if the system satisfies the Eigenstate Thermalization Hypothesis [3–5]. Furthermore, the entanglement entropy can detect when a system becomes many-body-localized, since in that case it scales again with the area of the boundary of the subregion [6].

If one is interested in distinguishing equilibrium from non-equilibrium situations, then in particular the mutual information can be useful. In equilibrium, the mutual information between two adjacent subregions of a Gibbs state with local interactions scales like the area between the subregions [7]; essentially because the two point correlations or coherences $G_{ij}$ between two lattice points $i$ and $j$ decay fast enough.

Out-of-equilibrium, this picture potentially differs. First studies of mutual information in non-equilibrium steady states (NESS) show a logarithm violation of the area law in free fermionic chains without noise [8], but an unmodified area law for the logarithmic negativity (an entanglement measure similar to the mutual information [9]) is found in a chain of quantum harmonic oscillators [10]. This behaviour can differ in the presence of a scattering region [11] (see comment[1]). The mutual information is also expected to follow an area law in driven integrable interacting systems in which transport is mainly ballistic (see [12] for a review). In contrast to this, a more recent study [13] finds (via numerical or approximate analysis) a volume law for the mutual information in two case studies: The non-interacting limit of a random unitary circuit and an Anderson tight-binding model. In both of these cases the system is diffusive, with a coherence length (the distance over which coherences are non-zero[2]) greater than the system size. In other words, the system is in the mesoscopic regime.[3]

---

[1]The emerging volume law for the mutual information between symmetric regions in this case has its origin in the entanglement between particles reflected and transmitted by the scattering region.

[2]Strong interactions in the presence of noise tend to decrease the coherence length.

[3]Mesoscopic systems are defined by the fact that the coherence length is greater than both the observation scale (to be able to observe quantum coherent effects) and the mean free path (in order to have diffusive transport).

We confirm the volume law for mutual information of entanglement in mesoscopic current driven systems through exact results for a model of noisy free fermions in its steady state (reached after long times): the quantum symmetric simple exclusion process (QSSEP) [14]. This model can be seen as a minimal description of coherent and diffusive transport in noisy mesoscopic systems [15] and has the advantage that it allows for analytical results without resorting to numerics. While the coherences $G_{ij}$ in QSSEP vanish under the noise average, the fluctuations of coherences survive in the steady state. We show that it is these non-vanishing fluctuations which are responsible for the volume law of the mutual information between two adjacent intervals of the system.[4] In doing so, we develop a mathematical method that allows to calculate the spectrum of a large class of random matrices from the information about its "local free cumulants" (see below for a definition). For QSSEP this is precisely the information about the fluctuations of coherences.

Interestingly, the mutual information of the classical simple symmetric exclusion process (SSEP) only satisfies an area law for the mutual information [16, 17]. Keeping in mind the generic long range correlations in classical non-equilibrium systems [18], this shows that the coherences in QSSEP represent a stronger form of correlations not present in the classical description.

## 2 Results on entanglement in QSSEP

The open quantum symmetric simple exclusion process (QSSEP) is a one-dimensional chain with $N$ sites occupied by spinless free fermions $c_j^\dagger$ with noisy hopping rates. The system is driven out-of-equilibrium by two boundary reservoirs at different particle densities, respectively $n_a$ and $n_b$. While the bulk of the system evolves unitarily but stochastically, the boundary sites are driven via a dissipative but deterministic Lindbladian,

$$\rho_t \to \rho_{t+dt} = e^{-idH_t} \rho_t e^{idH_t} + \mathcal{L}_{\text{bdry}}(\rho_t)\, dt\,. \tag{1}$$

The stochastic Hamiltonian increment is

$$dH_t := \sum_{j=1}^{N-1} c_{j+1}^\dagger c_j \, dW_t^j + c_j^\dagger c_{j+1} \, d\overline{W}_t^j\,, \tag{2}$$

where $dW_t^j$ are increments of complex Brownian motions (whose time derivative is a white noise) with variance $\mathbb{E}[dW_t^j d\overline{W}_t^k] = \delta^{j,k} dt$ and $\mathbb{E}$ the noise average. On the boundary sites, particles are injected with rate $\alpha$ and extracted with rate $\beta$ by $\mathcal{L}^+(\bullet) = c^\dagger \bullet c - \frac{1}{2}\{cc^\dagger, \bullet\}$ and $\mathcal{L}^-(\bullet) = c \bullet c^\dagger - \frac{1}{2}\{c^\dagger c, \bullet\}$. The complete driving becomes,

$$\mathcal{L}_{\text{bdry}} = \alpha_1 \mathcal{L}_1^+ + \beta_1 \mathcal{L}_1^- + \alpha_N \mathcal{L}_N^+ + \beta_N \mathcal{L}_N^-\,,$$

and rates relate to the left and right reservoir densities by $n_a = \frac{\alpha_1}{\alpha_1+\beta_1}$ and $n_b = \frac{\alpha_N}{\alpha_N+\beta_N}$.

The key quantity of interest is the matrix of coherences (two-point-function or Green's function) $G_{ij}(t) := \text{Tr}(\rho_t c_i^\dagger c_j)$ which contains all information about the system since the evolution of QSSEP preserves Gaussian fermionic states [14]. In this article we are only interested in the steady state distribution of coherences, which is unique regardless of the initial condition.

Introducing continuous variables $x = i/N \in [0,1]$ in the large $N$ limit, we consider the system to be defined on $[0,1]$ instead of the discrete lattice $\{1, \cdots, N\}$. The $q$-th Renyi entropy of a subset $I \subset [0,1]$ of length $\ell_I$ is

$$S_I^{(q)} := (1-q)^{-1} \log \text{Tr}(\rho_I^q)\,, \tag{3}$$

---

[4]One has to pay attention to the fact that the noise averaged entropy differs form the entropy of the noise averaged state, $\mathbb{E}[S(\rho)] \neq S(\mathbb{E}[\rho])$.

where $\rho_I$ is the system's density matrix reduced to the subset $I$. It can be expressed in terms of the density of eigenvalues $\lambda \in [0,1]$ of the matrix of coherences reduced to this subset $G_I := (G_{ij})_{ij \in I}$. Its intensive part is

$$s_I^{(q)} := \frac{S_I^{(q)}}{N} = \frac{\ell_I}{1-q} \int d\sigma_I(\lambda) \log[\lambda^q + (1-\lambda)^q], \tag{4}$$

with $d\sigma_I(\lambda)$ the normalized spectral density. In the limit $q \to 1$ one obtains the (intensive part of the) van Neumann entropy

$$s_I^{(1)} := -\ell_I \int d\sigma_I(\lambda) [\lambda \log(\lambda) + (1-\lambda) \log(1-\lambda)]. \tag{5}$$

Since the system is in a mixed state, the entanglement entropy of a subsystem is not a meaningful quantity. Instead, we consider the (intensive part) of the mutual information

$$i^{(q)}(I_1; I_2) := s_{I_1}^{(q)} + s_{I_2}^{(q)} - s_{I_1 \cup I_2}^{(q)}. \tag{6}$$

Our analytical results (derived below) for the 2nd mutual information between two adjacent intervals $I_1 = [0, c]$ and $I_2 = [c, 1]$ of the system as a function of the cut at $c \in [0, 1]$ is shown in Fig. 1. It perfectly agrees with the numerical simulation. The 2nd Renyi entropy has been chosen for convenience in order to compare with the numerical estimates in Ref. [13] and since the phenomenology in the steady state usually does not differ for higher Renyi entropies or for the van Neumann entropy [19]. Since the figure shows the intensive part of the mutual information one immediately recognizes that mutual information in QSSEP follows a volume law. We included the result for the non-interacting random unitary circuit from Ref. [13] that takes into account only the second order fluctuations of coherences, but not their higher moments.

However, we insist that the volume law scaling of the mutual information in QSSEP follows from the fact that the support of the spectra $d\sigma_{[0,c]}(\lambda)$ and $d\sigma_{[c,1]}(\lambda)$ (which we find below in Eq. (10)) is larger than the intervals $[0, c]$ and $[c, 1]$, such that the effective density $c\, d\sigma_{[0,c]}(\lambda) + (1-c)d\sigma_{[c,1]}(\lambda) - d\sigma_{[0,1]}(\lambda)$, over which one integrates in Eq. (6), cannot cancel to zero. This is nicely illustrated in Fig. 2. Therefore, all Renyi and van Neumann mutual informations scale as the volume.

We make a few remarks before describing the exact formula for the spectral densities: Firstly, the spectrum for a reflected interval is related by symmetry,

$$d\sigma_{[0,c]}(\lambda) = d\sigma_{[1-c,1]}(1-\lambda). \tag{7}$$

That is, the interval $[0, c]$ is equivalent to the interval $[1-c, 1]$ if we interchange the reservoir densities $n_a \leftrightarrow n_b$ and thus reverse the mean current, which is equivalent to the replacement $\lambda \to 1 - \lambda$. Secondly, the spectral density for generic reservoir densities $n_a, n_b$ can be obtained from the special case $n_a = 0, n_b = 1$, since, in law, the eigenvalues for the generic case are $n_a + (n_b - n_a)\lambda$, where $\lambda$ is distributed according to the special case. This also shows that in the equilibrium case where $n_a = n_b$, all eigenvalues are equal to $n_a$. That is, $d\sigma_{[c_1,c_2]}(\lambda) = \delta(\lambda - n_a)d\lambda$ is independent of the interval $[c_1, c_2]$, which implies that the mutual information in Eq. (6) is zero and that it no longer follows a volume law.[5] Based on this discussion we can discuss the generic out-of-equilibrium case by taking in the following $n_a = 0$ and $n_b = 1$ if not stated otherwise.

---

[5]In the equilibrium case, the leading term of the cumulants of $G_{ij}$ vanishes in the large $N$ limit, and one has to consider sub-leading terms in order to confirm an area law scaling of the mutual information.

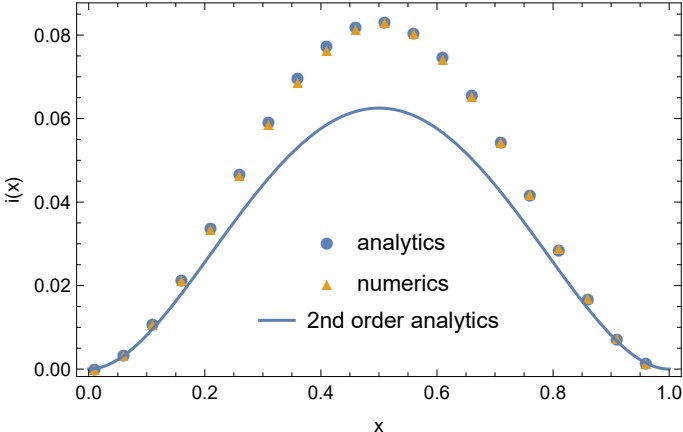

Figure 1: The intensive part $i(c) := i([0,c] : [c,1])$ of the 2nd Renyi mutual information as a function of the cut at $c$. The "analytical" data points are obtained from Eq. (8) via a numerical solution of Eq. (9). They differ quite substantially from the second order contribution based solely on $g_2(x,y)$, which shows that the higher order local free cumulants $g_{n\geq3}$ are important. This is compared to "numerical" data points from a numerical simulation of the QSSEP dynamics on $N = 100$ sites with discretization time step $dt = 0.1$. Instead of averaging over many noisy realizations, we exploit the ergodicity of QSSEP and perform a time average of a single realization between $t = 0.15$ and $t = 0.4$. The QSSEP dynamics reaches its steady state at approximately $t = 0.1$.

We find that the exact solution for the spectrum of $G_I$ reduced to the interval $I = [c,1]$ is (see appendix C)

$$d\sigma_{[c,1]}(\lambda) = \frac{d\lambda}{\pi\lambda(1-\lambda)}\frac{\theta}{\theta^2 + \log^2(re^{1/c})}\mathbb{I}_{\lambda\in[z_l(c),1]}. \tag{8}$$

Here, $\theta$ and $r$ are functions of $\lambda$ implicitly defined through the (transcendental) equations

$$1 + \log r = r\xi\cos\theta, \quad \theta = r\xi\sin\theta, \tag{9}$$

with $\xi = e^{1/c}(\frac{1-c}{c})(\frac{\lambda}{1-\lambda})$. The left boundary of the spectrum (the other being $\lambda = 1$) is

$$z_l(c) = \frac{c}{c + (1-c)e^{1/c}}. \tag{10}$$

In particular, the support of $d\sigma_{[c,1]}$ is larger than the interval $[c,1]$. Therefore the terms in Eq. (6) cannot compensate each other, such that the mutual information obeys the volume law.

Close to the right boundary, the spectral density $d\sigma_{[c,1]}$ approaches that of the complete interval, i.e. $d\sigma_{[c,1]}(\lambda) \simeq_{\lambda\to1} d\sigma_{[0,1]}(\lambda)$ (see below), while close to the left boundary, it vanishes as a square root, i.e. $d\sigma_{[c,1]}(\lambda) \simeq_{\lambda\to z_l(c)} \frac{c^2 d\lambda}{\pi[z_l(c)(1-z_l(c))]^{3/2}}\sqrt{2(\lambda - z_l(c))}$. This is similar to the Wigner semi-circle distribution for Gaussian random matrices.

The formula for $d\sigma_{[c,1]}$ simplifies considerably if we want to know the spectrum of the whole matrix $G$. In the limit $c \to 0$, we have $\xi \simeq \frac{e^{1/c}}{c}(\frac{\lambda}{1-\lambda}) \to \infty$, so that $re^{1/c} = \frac{1-\lambda}{\lambda}$, $\theta = \pi$ and we obtain

$$d\sigma_{[0,1]}(\lambda) = \frac{d\lambda}{\lambda(1-\lambda)}\frac{1}{\pi^2 + \log^2(\frac{1-\lambda}{\lambda})}\mathbb{I}_{\lambda\in[0,1]}. \tag{11}$$

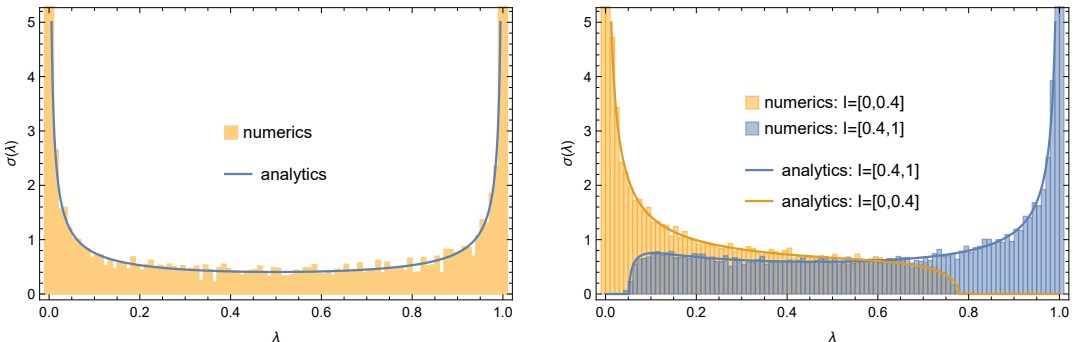

Figure 2: (Left) The spectral density $\sigma_I(\lambda)$ on the complete interval $I = [0,1]$. A comparison between the analytical prediction in Eq. (11) and a numerical simulation of $G$. The histogram of eigenvalues of $G$ corresponds to a single realization of the stochastic evolution of $G$. (Right) Spectral density $\sigma_I(\lambda)$ for the intervals $I = [0, 0.4]$ and $I = [0.4, 1]$. The support of the spectra is larger than the intervals $I$ and therefore the mutual information scales as the volume.

This is actually a Cauchy distribution

$$d\sigma_{[0,1]}(\nu) = \frac{d\nu}{\pi^2 + \nu^2}\,, \tag{12}$$

after the change of variables $\nu = \log(\frac{1-\lambda}{\lambda}) \in (-\infty, \infty)$. Thus, the spectral density of the complete density matrix corresponds to free fermions $\rho \propto \exp(\sum_j \nu_j d_{\nu_j}^\dagger d_{\nu_j})$, with pseudo energy densities $\{\nu_j\}$ given by the Cauchy distribution on the positive axis.

A comparison of the analytical expressions for the spectrum on different intervals with a numerical simulation is shown in Fig. 2.

Our method can be used to compute the mutual information for a variety of configurations. For instance, the von Neumann mutual information between two small distant intervals $I_1 = [c_1, c_1 + \ell_1]$ and $I_2 = [c_2, c_2 + \ell_2]$, is (for $c_1 < c_2$)

$$i^{(1)}(I_1; I_2) = \ell_1 \ell_2 \frac{2c_1(1-c_2)}{c_2 - c_1} \log \frac{(1-c_1)c_2}{(1-c_2)c_1} + O(\ell_1^2, \ell_2^2).$$

This echoes, once again, that quantum and classical correlations are long ranged out-of-equilibrium and implies the volume law for the mutual information.

## 3 Mathematical method

The expressions for the spectrum of QSSEP in its steady state follow from a quite general method that allows to determine the spectrum of sub-blocks of any random matrix $G$ satisfying three conditions (namely $U(1)$-invariance, scaling with matrix size $N$ and factorization) which are outlined in the next section. The necessary ingredient is a family of correlation functions

$$g_n(x_1, \cdots, x_n) := \lim_{N \to \infty} N^{n-1} \mathbb{E}[G_{i_1 i_2} G_{i_2 i_3} \cdots G_{i_n i_1}]^c\,. \tag{13}$$

Here we alternate between continuous arguments $x$ and matrix indices $i$ via $x = i/N \in [0,1]$. Note that the order of indices in Eq. (13) forms a loop. In analogy to free cumulants in free probability theory (see below), we call these functions "local free cumulants".[6] They fully

---

[6] They are local in the sense that they depend on the internal structure of $G$. This is different to most of the usual random matrix ensembles (e.g. GUE or Haar randomly rotated matrices) for which the correlation functions $g_n$ would be independent of the position $x$.

determine the probability measure on the matrix $G$.

For the slightly more general aim of finding the spectrum of $G_h := h^{1/2}Gh^{1/2}$, with $h$ a diagonal matrix, we consider the functional

$$F[h](z) := \mathbb{E}\,\underline{\mathrm{tr}}\log(z - G_h),\tag{14}$$

with $\underline{\mathrm{tr}} = \mathrm{tr}/N$ the normalized $N$-dimensional trace. Its derivative is the resolvent $R[h](z) := \mathbb{E}\,\underline{\mathrm{tr}}(z - G_h)^{-1}$ which contains information about the spectrum of $G_h$. In the special case where $h(x) = \mathbb{I}_{x \in I}$ is the indicator function on an interval (or on unions of intervals) we recover the spectral density of $G_I$ from the resolvent $R_I$ as $R_I(\lambda - i\epsilon) - R_I(\lambda + i\epsilon) = 2i\pi\ell_I d\sigma_I(\lambda)$, which is what we are interested in.

Our main result is that $F[h](z)$ is determined by the variational principle

$$\min_{a_z, b_z}\left[\int\left[\log(z - h(x)b_z(x)) + a_z(x)b_z(x)\right]dx - F_0[a_z]\right],\tag{15}$$

where the information specific to the random matrix ensemble we consider is contained in the following generating function (with $\vec{x} = (x_1, \cdots, x_n)$)

$$F_0[p] := \sum_{n \geq 1}\frac{1}{n}\int(\prod_{k=1}^{n}dx_k p(x_k))\,g_n(\vec{x}).\tag{16}$$

The extremization conditions read

$$a_z(x) = \frac{h(x)}{z - h(x)b_z(x)}, \quad b_z(x) = R_0[a_z](x),\tag{17}$$

with $R_0[a_z](x) := \frac{\delta F_0[a_z]}{\delta a_z(x)}$. Alternatively, Eqs.(17) read $zh(x)^{-1} = R_0[a_z](x) + a_z(x)^{-1}$, which resembles a local version of the so-called R-transform of free probability theory [20–23] (hence the name "local free cumulants" in Eq. (13)). A solution for $b_z$ determines the resolvent via

$$R[h](z) = \int\frac{dx}{z - h(x)b_z(x)},\tag{18}$$

and thereby the spectral density.

In the case of QSSEP, it is known [15, 24] that the local free cumulants $g_n$ are free cumulants w.r.t. the Lebesgue measure on the interval $[0, 1]$. We can thus use techniques from free probability (notably the relation between the R- and Cauchy transform) to show that $b_z$ satisfies the differential equations (see appendix B)

$$zb''(x) + h(x)(b'(x)^2 - b(x)b''(x)) = 0.\tag{19}$$

For $h(x) = \mathbb{I}_{x \in I}$, this reduces to

$$\begin{cases}[\log(z - b_z(x))]'' = 0, & \text{if } x \in I,\\ b_z(x)'' = 0, & \text{if } x \notin I,\end{cases}\tag{20}$$

with boundary conditions $b_z(0) = 0$ and $b_z(1) = 1$. For $I = I_1 \cup I_2 \cdots \cup I_k$, the union of intervals, an ansatz for a solution inside the intervals is $b_z(x) = z + u_j e^{v_j x}$, while outside the intervals $b_z$ is $x$-linear. The linearity coefficients and the coefficients $u_j$, $v_j$ are determined by imposing continuity of $b_z$ and of its first derivative. The resolvent is reconstructed using Eq. (18), and the spectral density $d\sigma_I$ by extracting the discontinuity of the resolvent on its cuts. Solving these equations for $I = [c, 1]$ leads to (8,9).

**Essentially classical.** Note that $F_0$ depends only on a small part of the information contained in the QSSEP correlation functions $g_n(\vec{x})$. Indeed, due to the integration in Eq. (16), the generating function $F_0$ depends only on a symmetrized version of $g_n(\vec{x})$ which is invariant under permutation of its arguments, i.e. on $\sum_{\sigma \in S_n} g_n(\sigma(\vec{x}))$ with $\sigma(\vec{x}) := (x_{\sigma(1)}, \cdots, x_{\sigma(N)})$ and $S_n$ the symmetric group of order $n$. Through the relation $\langle e^{\sum_i h_i \tau_i} \rangle_{\text{SSEP}} = \mathbb{E}[\text{Tr}(\rho\, e^{\sum_i h_i c_i^\dagger c_i})]$, which relates the classical SSEP (with particle density $\tau_i$ on site $i$) and the QSSEP, one can show (see Eq. (17) in [14] or section 4.2 in [25]) that this information is already contained in the classical SSEP. That is, the connected density-correlation functions of the SSEP is given as sum of the QSSEP correlation function of coherences $g_n(\vec{x})$ with permuted arguments,

$$\langle \tau_{i_1} ... \tau_{i_n} \rangle^c_{\text{SSEP}} = (-1/N)^{n-1} \frac{1}{n} \sum_{\sigma \in S_n} g_n(\sigma(\vec{x})),$$

where continuous positions and discrete lattice sites are related by $x = i/N$. Whether this phenomenon is specific to QSSEP or is also valid for more general chaotic mesoscopic systems is an open question.

# 4 Sketch of a proof

We will explain this method and further applications in greater detail in a mathematical follow paper [26]. Our method works if, in the limit of large matrix size $N$, the measure $\mathbb{E}$ of the random matrix $G$ satisfies three properties (see [15] section II.A for more details):

(i) local $U(1)$-invariance: In distribution, $G_{ij} \overset{d}{=} e^{-i\theta_i} G_{ij} e^{i\theta_j}$ for any angles $\theta_i$ and $\theta_j$

(ii) "Loop" without repeated indices scale as $\mathbb{E}[G_{i_1 i_2} G_{i_2, i_3} \ldots G_{i_n i_1}] = \mathcal{O}(N^{1-n})$

(iii) Factorization of the measure at leading order for products of "loops". Even if $i_1 = j_1$ is repeated, $\mathbb{E}[G_{i_1 i_2} \ldots G_{i_m i_1} G_{j_1 j_2} \ldots G_{j_n j_1}] = \mathbb{E}[G_{i_1 i_2} \ldots G_{i_m i_1}] \mathbb{E}[G_{j_1 j_2} \ldots G_{j_n j_1}]$

These conditions ensure that the moments of $G$ can be expressed as a sum over non-crossing partitions of the set $\{1, \cdots, n\}$ (see Eq. (30) in [15])

$$\phi_n := \mathbb{E}\underline{\text{tr}}(G^n) = \sum_{\pi \in NC(n)} \int g_{\pi^*}(\vec{x})\, \delta_\pi(\vec{x}) d\vec{x}, \tag{21}$$

where $g_\pi(x) := \prod_{b \in \pi} g_{|b|}(\vec{x}_b)$ with $\vec{x}_b = (x_i)_{i \in b}$. With $\delta_\pi(\vec{x})$ we mean a product of delta functions that equate all $x_i$ with $i$ in the same block $b \in \pi$ and $\pi^*$ is the Kreweras complement of $\pi$ (see below for an example and e.g. [21] for a set of lecture notes).

Similarly, the moments $\phi_n[h] := \mathbb{E}\underline{\text{tr}}(G_h^n)$ can be expressed as a sum over non-crossing partitions, if above we replace $g_{\pi^*}(\vec{x})$ with $g_{\pi^*}(\vec{x})h(x_1)\cdots h(x_n)$. To better understand the structure of this sum, we note that non-crossing partitions $\pi \in NC(n)$ are in one-to-one correspondence with planar bipartite rooted trees with $n$ edges. Here is an example for $\pi = \{\{1,3\},\{2\},\{4,5\},\{6\}\}$ (dotted lines) whose Kreweras complement is $\pi^* = \{\{\bar{1},\bar{2}\},\{\bar{3},\bar{5},\bar{6}\},\{\bar{4}\}\}$ (solid lines).

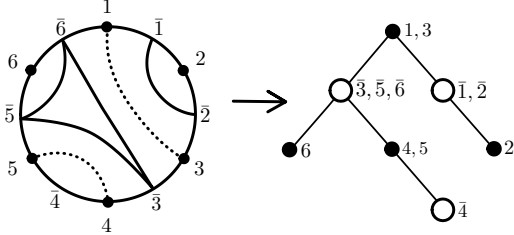

The blocks $b$ of the partition $\pi$ are associated with black vertices and the blocks $w$ of the Kreweras complement $\pi^*$ with white vertices. They are connected if they have an element in common. The root is (by convention) chosen to be the block $b$ containing 1.

To apply this correspondence to our problem, we define $\phi_n[h](x) = \mathbb{E}\langle x|(G_h)^n|x\rangle$ with $\phi_n[h] = \int \phi_n[h](x)dx$ such that we can associate the root of the tree to the marked point $x$. Then $z^{-n}\phi_n[h](x) = \sum_{T_\bullet} W(T_\bullet^x)$ is equal to a sum over all bipartite trees $T_\bullet$ rooted at a black vertex with label $x$ and with $n$ edges in total. Their weight $W(T_\bullet^x)$ is defined as follows: A black vertex carries the integration variables $x_i$, the root carries $x$. A white vertex whose neighbours are $x_1, \cdots, x_k$ takes the value $z^{-k}h(x_1)\cdots h(x_k)g_k(x_1, \cdots, x_k)$. Finally one takes the product over all vertices and integrates over all $x_i$ (except for the root $x$). By definition we set the tree consisting of a root without legs to one. Graphically the rules for the weights $W(T_\bullet^x)$ are:

$$\text{[white vertex diagram]} = z^{-k}h(x_1)\cdots h(x_k)g_k(x_1, \cdots, x_k)$$

$$\text{[black vertex diagram]} \quad x_i \quad = \int_0^1 dx_i$$

As an analogue of the resolvent, but with a marked point, we define $a_z(x) := \mathbb{E}\langle x|\frac{h}{z-G_h}|x\rangle = \frac{h(x)}{z}\sum_{n\geq 0}\frac{\phi_n[h](x)}{z^n}$. The correspondence with trees implies that

$$a_z(x) = \frac{h(x)}{z}\sum_{T_\bullet} W(T_\bullet^x). \tag{22}$$

Among these trees, we denote by $T_\circ$ those whose root has a single leg only. This defines

$$b_z(x) = \frac{z}{h(x)}\sum_{T_\circ} W(T_\circ^x). \tag{23}$$

The trees $T_\circ$ can be understood as the "no $x$" terms in the expansion of $\mathbb{E}\langle x|(G_h)^n|x\rangle$, i.e. those terms where none of the intermediate indices is equal to $i$ with $x = i/N$. In this sense, $b_z(x) = \frac{z}{h(x)}\mathbb{E}\langle x|\frac{G_h}{z-G_h}|x\rangle^{[\text{no}\,x]}$.[7]

Since any tree $T_\bullet$ with $l$ legs on the root can be written as a product of $l$ trees of the form $T_\circ$ we have $\sum_{T_\bullet} W(T_\bullet^x) = (1 - \sum_{T_\circ} W(T_\circ^x))^{-1}$ and this implies the first relation in Eq. (17). For the second relation, we start with $T_\circ$ and cut the $l$ outgoing legs of the first white vertex. This generates a product of $l$ trees $T_\bullet$. Then $\sum_{T_\circ} W(T_\circ^x)$ is equal to

$$\frac{h(x)}{z}\sum_{l\geq 0}\int\left(\prod_{i=1}^l dx_i\frac{h(x_i)}{z}\sum_{T_\bullet} W(T_\bullet^{x_i})\right)g_{l+1}(x, x_1, \cdots, x_l), \tag{24}$$

which implies the second relation. Finally, expanding $F[h](z) = \log(z) - \sum_{n\geq 1}\frac{z^{-n}}{n}\phi_n[h]$, one sees that

$$-h(x)\frac{\delta F[h](z)}{\delta h(x)} = \sum_{T_\bullet} W(T_\bullet^x) - 1 = a_z(x)b_z(x), \tag{25}$$

which justifies the variational principle in Eq. (15).

---

[7]Using this way of writing $b_z(x)$ opens the route for an yet alternative proof of our main result [26].

# 5 Conclusion

We have analyzed the mutual information in the open QSSEP, determined exact formulae in various configurations and proved that it obeys a volume law. This reflects the presence of long range quantum correlations in out-of-equilibrium mesoscopic systems. The derivation of these results is based on a new method that allows to evaluate the typical spectrum of sub-blocks of structured random matrices. Here we applied it to QSSEP but its scope is likely to be wider. For instance, the variational principle in Eq. (15) is very similar to the one for the free energy of Bernoulli variables [25]. It also has potential applications in many-body systems at equilibrium satisfying the ETH. Indeed, as recently understood [27, 28], a proper formulation of this hypothesis involves notions from free probability. We showed in appendix A how to evaluate, within ETH, canonical or micro-canonical expectation values of observables in terms of their local free cumulants.

Our analysis leads to the question: for which class of noisy driven systems does the mutual informations satisfies a volume law? Clearly those systems have to be in the mesoscopic regime – with a coherence length greater than the observation scale. Does this class coincide with that discussed in [15] whose coherent fluctuations show free probability structures? The fact that the Anderson model has similar entanglement properties [13] hints that a large variety of metallic materials should have entanglement properties similar to the QSSEP universality class.

Furthermore, we observed that the entanglement properties of QSSEP depend only on a small part of the data available in the distribution of coherences $g_n(\vec{x})$. In fact, although mutual information in QSSEP differs from that in SSEP, the necessary data is indirectly already contained in its classical version. One may therefore wonder, whether this relation between transport properties (diffusion constant, mobility, i.e. classical properties) and entanglement properties (quantum properties) extends to more general chaotic and diffusive quantum systems. That is, we wonder if the necessary information for entanglement properties is already hidden in the (classical) macroscopic fluctuation theory (MFT) [29], or if not, what extra minimal information is needed. In other words, we may introduce two notions of universality classes for quantum diffusive chaotic systems: (i) universality for transport properties, this would be related to classical universality classes, and (ii) universality for coherent or entanglement properties, this would be related to quantum universality classes. Whether these two notions of universalities coincide or differ is still an open question, and the answer to this question might depend on which properties we include in the specification of the quantum universality classes.

Note that our analysis in this paper deals with the typical or mean entanglement entropy. However, fluctuations of entropies are known in the case of the closed QSSEP (periodic system without boundary driving) [30] and it would be interesting to decipher how this extends to the out-of-equilibrium setting discussed here.

Finally, it would be interesting to find a protocol for information transfer in QSSEP and link our results to its capacity as a quantum channel.

## Acknowledgments

We thank Lorenzo Piroli for earlier collaboration on this topic, and Ph. Biane and A. Nahum for discussions.

**Funding information** This work was in part supported by the CNRS, the ENS and the ANR project ESQuisses under contract number ANR-20-CE47-0014-01.

# A  Application to the eigenstate thermalization hypothesis (ETH)

The variational principle described in the main text provides an efficient way to handle the structure and consequences of contact points in say multiple products of random matrices. As such it potentially has a larger domain of applicability. For instance, it also appears in the classical problem of dealing with the free energy of Bernoulli variables [25]. It also has potential applications in quantum many-body physics at equilibrium via the eigenstate thermalization hypothesis [3–5], as we now explain.

Let us consider a closed macroscopic system with $\mathcal{H}$ its Hilbert space, whose dimension $d_H$ is exponentially large in the system size or volume, and $H$ its hamiltonian. Let $|E_i\rangle$ be the energy eigenstates, $H|E_i\rangle = E_i|E_i\rangle$, which we assume to be non degenerate for simplicity. In a large but bounded volume, the energy spectrum is discrete, and the splitting between two successive energies is exponentially small with the system size. The density of state $\sigma(E)dE$, that is the number of eigenenergy is the interval $[E, E+dE]$, is thus exponential in the system size. By Boltzmann formula, $\sigma(E)dE = e^{S(E)}dE$ with $S(E)$ the entropy of the total system at energy $E$ ($S(E)$ is extensive in the system size). We normalized it so that its total integral in one. The inverse temperature, at energy $E$, is defined as $\beta_E = S'(E)$, where the prime denotes the derivative.

The eigenstate thermalisation hypothesis (ETH) is a statement about the structure of the matrix elements of physical observables in the energy eigenbasis. It asserts that the matrix elements of say local operators $A$, in the energy eigenstates $|E_i\rangle$ in the bulk of the spectrum, take the following form [3–5]

$$A_{ij} = \mathcal{A}(\bar{E}_{ij})\delta_{ij} + e^{-\frac{1}{2}S(\bar{E}_{ij})}R^A_{ij}f_A(\bar{E}_{ij}, \omega_{ij}),\tag{A.1}$$

with $\bar{E}_{ij} = \frac{1}{2}(E_i + E_j)$, $\omega_{ij} = E_i - E_j$, and $\mathcal{A}(\bar{E})$ and $f_A(\bar{E}, \omega)$ smooth functions of their arguments, with $f_A$ rapidly decreasing with $\omega$, and $R^A_{ij}$ matrices of order one with erratically varying elements in the range of energy around $\bar{E}$, with zero mean and unit variance (w.r.t. some measure to be discussed below).

Schematically, ETH implies that the matrix $A_{ij}$ can approximatively be thought of as a (random) band matrix whose width is determined by the decay rate of $f_A$ as a function of $\omega$. One can show [3–5, 31] that the decay rate at energy $E$ is the temperature $1/\beta_E$, the natural energy scale at energy $E$. Furthermore, the function $f_A$ is expected to be approximatively constant on an energy scale of the order of the Thouless energy $\mathcal{E}_T \simeq \hbar D/L^2$, the inverse of the diffusion time, with $D$ the diffusion constant and $L$ the linear size of the system. Since $\mathcal{E}_T$ decreases only polynomially with $1/L$, not exponentially with $L$, there is an exponentially large number of states in any energy window of size $\mathcal{E}_T$. The measure mentioned above, w.r.t. which $R^A_{ij}$ has zero mean and unit variance, is the one obtained by sampling on an energy window of size the Thouless energy $\mathcal{E}_T$.

The ETH, together with the statement that $R^A_{ij}$ has zero mean and unit variance, ensures that the one and two-point expectations, say in wave packets centred around an energy $E$, coincide with the thermodynamic correlations at the inverse temperature $\beta_E$ [3–5, 31].

However, it has recently been understood [27, 28] that the original formulation of ETH needs to be generalized in order to be able to deal with multi-time correlation functions of say collections of operators $A_\alpha$ at different times $t_\alpha$: namely, $\text{tr}(\rho_\beta A_1(t_1)\cdots A_n(t_n))$, with $\rho_\beta = Z^{-1}e^{-\beta H}$ the equilibrium Gibbs states at the inverse temperature $\beta = 1/k_B T$ and $Z$ the partition function $Z = \text{tr}(e^{-\beta H})$ where the trace is over the system Hilbert space. The generalized ETH [27, 28] asserts that the expectation (w.r.t. to the sampling measure on energy window of width the Thouless energy which we call the ETH measure) of products of matrix elements of local operators $A_\alpha$ or $B_\alpha$ in the eigenenergy basis is such that, for all distinct indices

$i_k$ and $j_k$,

$$\overline{(A_1)_{i_1 i_2}(A_2)_{i_2 i_3}\cdots(A_n)_{i_n i_1}} = e^{-(n-1)S(\bar{E}_{12\cdots n})}\kappa_n(E_1, E_2, \cdots, E_n), \qquad \text{(A.2a)}$$

$$\overline{(A_1)_{i_1 i_2}\cdots(A_n)_{i_n i_1}(B_1)_{j_1 j_2}\cdots(B_m)_{j_m j_1}} = \overline{(A_1)_{i_1 i_2}\cdots(A_n)_{i_n i_1}}\cdot\overline{(B_1)_{j_1 j_2}\cdots(B_m)_{j_m j_1}}, \qquad \text{(A.2b)}$$

with $E_k$ the energy of the eigenstate $|E_{i_k}\rangle$ and $\bar{E}_{12\cdots n} = \frac{1}{n}(E_1 + E_2 + \cdots + E_n)$ the mean energy. Furthermore, the ETH expectations of product of matrix elements for which the set of in-going indices is not a permutation of the set of out-going indices vanish. These rules are enough to compute the expectation of any product of matrix elements since any permutation can be decomposed into product of cycles. Of course, these properties directly translate to those of the ETH expectations of the time translated operators $A_\alpha(t_\alpha)$ since $A_\alpha(t_\alpha) = e^{+iHt_\alpha}A_\alpha e^{-iHt_\alpha}$ have simple matrix elements in the energy eigenbasis.

Note that the indices appearing in Eq. (A.2a) form a loop, so that we can refer to such expectations as loop expectations. The functions $\kappa_n(E_1, E_2, \cdots, E_n)$ has been understood as free cumulants in [28]. It is easy to verify that those ETH expectation values satisfy the three criteria – $U(1)$ invariance, scaling and factorisation – necessary for the free probability techniques to be emerging [15]. Of course, the $\kappa_n$'s depend on the operators $A_\alpha$. In the above notation $\kappa_1(E) = \mathcal{A}(E)$. Equation (A.2b) indicates that the ETH expectation of products of loops factorizes into products of expectations.

As it was understand in [15,28], the structure of the generalised ETH and that of coherence fluctuations in mesoscopic systems, and in particular in Q-SSEP, has a similar origin: the coarse-graining at microscopic either spatial or energy scales, and the unitary invariance at these microscopic scales.

When computing multi-point thermal expectations, such as $\text{tr}(\rho_\beta A_1(t_1)\cdots A_n(t_n))$, one has to sum over intermediate energy eigenstates. That is: one has to sum over multiple indices $i_k$ labelling complete sets of energy eigenstates. These sums cannot be directly represented as integral over the energy spectrum, with density $e^{S(E)}dE$, because the matrix elements of the operators $A_\alpha$ vary erratically. Thus, one has first to do a local sum, or a averaged smearing, by sampling the eigenstates on a energy window of size $\mathcal{E}_T$ around a given energy – and this leads to the ETH averages which are smooth as functions of the energies – and, in a second step, one then represents the sum over all these local averages by an integral over the energy spectrum. Symbolically, the rules to compute correlation functions in ETH framework is (with $d_{\text{H}}$ the Hilbert space dimension).

$$d_{\text{H}}^{-1}\sum_i (\cdots) \rightsquigarrow \int dE\, e^{S(E)}\,\overline{\text{ETH.average}(....)}. \qquad \text{(A.3)}$$

For instance, $\text{tr}(\rho_\beta A) = Z^{-1}\sum_i e^{-\beta E_i}A_{ii} = Z^{-1}\int dE\, e^{S(E)-\beta E}\kappa_1(E)$ and $Z = \int dE\, e^{S(E)-\beta E}$, which can evaluated via a saddle point as usual, while

$$\text{tr}(\rho_\beta AB) = Z^{-1}\sum_{ij} e^{-\beta E_i}A_{ij}B_{ji}$$

$$= Z^{-1}\left[\int dE_1 dE_2\, e^{S(E_1)-\beta E_1}e^{S(E_2)-S(\bar{E}_{12})}\kappa_2(E_1, E_2) + \int dE_1\, e^{S(E_1)-\beta E_1}\kappa_1(E_1)^2\right].$$

To go from the first to the second line, we have splitted the sum in two sub-sums, one in which $i \neq j$ and the other with $i = j$, and then applied the ETH rules. For higher order correlation functions this splitting procedure becomes more and more cumbersome. But this is what the variational principle handles efficiently.

In general, one may evaluate the canonical expectation values $\text{tr}(\rho_\beta A_1(t_1)\cdots A_n(t_n))$ or, if appropriately regularized, simply $\underline{\text{tr}}(A_1(t_1)\cdots A_n(t_n))$ with $\underline{\text{tr}}$ the normalised trace,

$\underline{\mathrm{tr}}(\cdots) := d_{\mathrm{H}}^{-1}\mathrm{tr}(\cdots)$ and $d_{\mathrm{H}}$ the dimension of the system Hilbert space. Alternatively, we may compute the micro-canonical expectation values $\langle E|A_1(t_1)\cdots A_n(t_n)|E\rangle$, for a given energy eigenstate (with extensive energy). By standard arguments, these will be equivalent to the canonical Gibbs expectation values at inverse temperature $\beta_E$.

To handle all possible expectations for all considered operators $A_\alpha(t_\alpha)$, let us introduce a formal free algebra with generators $\mathfrak{a}_\alpha$ and the formal series $\mathbb{A}$ defined as

$$\mathbb{A} := \sum_\alpha A_\alpha(t_\alpha)\,\mathfrak{a}_\alpha\,. \tag{A.4}$$

The benefit of this formal manipulation is that we now have to deal with only one object, namely $\mathbb{A}$, to handle the collection of all operators. Of course $\mathbb{A}$ takes values in a large formal free algebra. To know about the thermal expectations $\mathrm{tr}(\rho_\beta A_1(t_1)\cdots A_n(t_n))$, we have to evaluate the expectations $\mathrm{tr}(\rho_\beta \mathbb{A}^n)$, or more generally,

$$\mathrm{tr}\big(\rho_\beta \mathbb{A}_h^n\big) \text{ or } \underline{\mathrm{tr}}\big(\mathbb{A}_h^n\big), \text{ with } \mathbb{A}_h = h^{1/2}\,\mathbb{A}\,h^{1/2}\,, \tag{A.5}$$

with $h$ a matrix, diagonal in the energy basis, with smoothly varying diagonal matrix element $h(E)$. Alternatively, we may deal with the micro-canonical expectation values

$$\langle E|\mathbb{A}_h^n|E\rangle\,. \tag{A.6}$$

Expanding $\mathrm{tr}(\rho_\beta \mathbb{A}_h^n)$ or $\langle E|\mathbb{A}_h^n|E\rangle$ and picking the word $\mathfrak{a}_1\mathfrak{a}_2\cdots\mathfrak{a}_n$ yields the expectations $\mathrm{tr}(\rho_\beta A_1(t_1)\cdots A_n(t_n))$ or $\langle E|A_1(t_1)\cdots A_n(t_n)|E\rangle$, respectively.

The problem is now to evaluate the expectations $\mathrm{tr}(\rho_\beta \mathbb{A}_h^n)$, $\underline{\mathrm{tr}}(\mathbb{A}_h^n)$ or $\langle E|\mathbb{A}_h^n|E\rangle$ using the ETH rules given, with the free cumulants of $\mathbb{A}$ as input data:

$$\overline{\mathbb{A}_{i_1 i_2}\mathbb{A}_{i_2 i_3}\cdots\mathbb{A}_{i_n i_1}} = e^{-(n-1)S(\bar{E}_{12\cdots n})}\,\kappa_n^{\mathbb{A}}(E_1, E_2, \cdots, E_n)\,. \tag{A.7}$$

This is exactly the problem the variational principle answers. Of course, the free cumulants $\kappa_n^{\mathbb{A}}$ depend on the set of operators $A_\alpha$, on the times $t_\alpha$, and the free algebra elements $\mathfrak{a}_\alpha$. Again expanding it in words in $\mathfrak{a}_\alpha$ yields the mutual free cumulants of the operators $A_\alpha$.

As in the main text, we define two generating function $a_z(E)$ and $b_z(E)$ by

$$a_z(E) := \langle E|\frac{h}{z - \mathbb{A}_h}|E\rangle = h(E)z^{-1}\sum_{n\geq 0}z^{-n}\langle E|\mathbb{A}_h^n|E\rangle\,, \tag{A.8}$$

$$b_z(E) := \langle E|\frac{z\mathbb{A}_h}{h(z - \mathbb{A}_h)}|E\rangle^{[\mathrm{no}\,E]} = h(E)^{-1}z\sum_{n\geq 1}z^{-n}\langle E|\mathbb{A}_h^n|E\rangle^{[\mathrm{no}\,E]}\,. \tag{A.9}$$

Here "$[\mathrm{no}\,E]$" means that the eigenstate $|E\rangle$, with energy $E$, does not appear in any of intermediate states (or resolution of the identity) used to evaluate the matrix elements $\langle E|\mathbb{A}_h^n|E\rangle$. By construction, the generating function for the multiple traces $\underline{\mathrm{tr}}\big(\mathbb{A}_h^n\big)$ can be written in terms of the function $a_z(E)$

$$\underline{\mathrm{tr}}\left(\frac{1}{z - \mathbb{A}_h}\right) = \int dE e^{S(E)}\,a_z(E)h(E)^{-1}\,. \tag{A.10}$$

Alternatively, one may compute directly the canonical Gibbs expectation values via

$$Z^{-1}\mathrm{tr}\big(e^{-\beta H}(z - \mathbb{A}_h)^{-1}\big) = Z^{-1}\int dE e^{S(E)-\beta E}\,a_z(E)h(E)^{-1}\,. \tag{A.11}$$

As usual, the later can be evaluated via a saddle point method reducing it to the micro-canonical expectation values at the energy corresponding to the inverse temperature $\beta$ (i.e. $E$ such that $\beta_E = \beta$).

Following the same reasoning as in the main text (see [26] for more details), it is then easy to prove that

$$a_z(E) = \frac{h(E)}{z - h(E) b_z(E)}, \tag{A.12a}$$

$$b_z(E) = \mathbb{R}_0[a_z](E), \tag{A.12b}$$

with

$$\mathbb{R}_0[a_z](E_0) := \sum_{n \geq 0} \int \left( \prod_{i=1}^n dE_i e^{S(E_i)} \right) e^{-nS(\bar{E}_{01\cdots n})} \kappa_{n+1}^{\mathbb{A}}(E_0, E_1, \cdots, E_n) a_z(E_1) \cdots a_z(E_n), \tag{A.13}$$

where $\bar{E}_{01\cdots n}$ is the mean energy, $\bar{E}_{01\cdots n} = \frac{1}{n+1}(E_0 + E_1 + \cdots + E_n)$. Equation (A.12a) is a consequence of the factorisation property (A.2). Equation (A.12b) follows from developing $b(z; E)$ in free cumulants and organising the sum according to the dimension of the block to which $E$ belongs to.

Since $\kappa_{n+1}^{\mathbb{A}}$ are rapidly decreasing as function of $\omega_i := E_i - \bar{E}_{01\cdots n}$, the difference of energy w.r.t. the mean energy, we can expand the entropy $S(E_i)$ as $S(E_i) = S(\bar{E}_{01\cdots n}) + S'(\bar{E}_{01\cdots n})\omega_i + \frac{1}{2}S''(\bar{E}_{01\cdots n})\omega_i^2 + \cdots$. The first derivative of the entropy is the temperature at the mean energy, $S'(\bar{E}_{01\cdots n}) = \beta_{\bar{E}_{01\cdots n}}$. The second derivative is proportional to the inverse of the heat capacity and scales as the inverse of the volume of the system. It can thus be neglected if the free cumulants $\kappa_{n+1}^{\mathbb{A}}$ decrease fast enough as functions of the $\omega_i$. Hence, $\sum_i S(E_i) - nS(\bar{E}_{01\cdots n}) = \beta_{\bar{E}_{01\cdots n}}(\bar{E}_{01\cdots n} - E_0) + \cdots$, and we can alternatively write $\mathbb{R}[\hat{a}](E_0)$ as

$$\mathbb{R}_0[a_z](E_0) = \sum_{n \geq 0} \int \prod_{k=1}^n dE_k\, e^{-\beta_{\bar{E}_{01\cdots n}}(E_0 - \bar{E}_{01\cdots n})} \kappa_{n+1}^{\mathbb{A}}(E_0, E_1, \cdots, E_n) a_z(E_1) \cdots a_z(E_n). \tag{A.14}$$

In some cases, approximating $\beta_{\bar{E}_{01\cdots n}}$ by $\beta_{E_0}$ could be a good further approximation.

The first few terms of the generating functions $a_z(E)$ and $b_z(E)$, in the simplest case with $h(E) = 1$, read

$$a_z(E_0) = z^{-1} + z^{-2} \kappa_1^{\mathbb{A}}(E_0) + z^{-3}\left[ \kappa_1^{\mathbb{A}}(E_0)^2 + \int dE_1 e^{S(E_1) - S(\bar{E}_{02})} \kappa_2^{\mathbb{A}}(E_0, E_1) \right] + \cdots,$$

$$b_z(E_0) = \kappa_1^{\mathbb{A}}(E_0) + \int dE_1 e^{S(E_1) - S(\bar{E}_{01})} \kappa_2^{\mathbb{A}}(E_0, E_1)\left[ z^{-1} + z^{-2}\kappa_1^{\mathbb{A}}(E_1) + \cdots \right]$$

$$+ z^{-2} \int dE_1 dE_2 e^{S(E_1) + S(E_2) - S(\bar{E}_{012})} \kappa_3^{\mathbb{A}}(E_0, E_1, E_2) + \cdots$$

Of course, similar simplifications as those done in Eq. (A.14) can be applied here.

Equations (A.10) or (A.11) supplemented to (A.12a), (A.12b) and (A.14), thus expresses the generating function of thermal expectation values in terms of the free cumulants. If we view those free cumulants as a minimal set of data coding for the information relative to all multi-point expectations, the equations (A.12a), (A.12b) and (A.10) encode the transformation from this minimal data set to the physically relevant quantities, namely the multi-point expectation values of local operators. If an analogy is needed, this is similar to the known fact in statistical or quantum field theory that the one-particle irreducible diagrams form a minimal complete data set and that the generating function of the connected correlation functions is the Legendre transform of the effective action which is the generating function of the one-particle irreducible diagrams. Here, the transformation from the free cumulants to the thermal expectation values is not a Legendre transformation but the variational principle formulated in the main text.

# B  Formula for $F_0[p]$ and equation for $b_z(x)$ in QSSEP

Here we derive an explicit expression for $F_0$ from Eq. (16) for QSSEP and use it to derive a differential equation for $b_z(x)$ in Eq. (20).

Recall the definition of the generating function $F_0[p] := \sum_{n \geq 1} \frac{1}{n} \int d\vec{x}\, p(x_1)...p(x_n) g_n(\vec{x})$. Defining $\mathbb{I}_{[p]}(y) := \int_y^1 dx\, p(x)$, we shall prove that for QSSEP,

$$F_0[p] = w - 1 - \int_0^1 dx \log[w - \mathbb{I}_{[p]}(x)], \qquad \text{with} \qquad \int_0^1 \frac{dx}{w - \mathbb{I}_{[p]}(x)} = 1. \qquad \text{(B.1)}$$

The QSSEP correlation functions $g_n$ are recursively given by (see Eq. (76) in [15])

$$\sum_{\pi \in NC(n)} g_\pi(\vec{x}) = \min(\vec{x}) =: \varphi_n(\vec{x}), \qquad \text{(B.2)}$$

with $g_\pi(x) := \prod_{b \in \pi} g_{|b|}(\vec{x}_b)$ and $\vec{x}_b = (x_i)_{i \in b}$. We view them as the free cumulants of indicator functions $\mathbb{I}_x(y) = 1_{y<x}$ with respect to the Lebesgue measure $d\mu(y) = dy$, since the moments of these functions are precisely $\mathbb{E}[\mathbb{I}_{x_1}...\mathbb{I}_{x_n}] = \min(\vec{x})$.

Integrating Eq. (B.2) we have $\varphi_n[p] = \sum_{\pi \in NC(n)} g_\pi[p]$, with $\varphi_n[p] = \int d\vec{x}\, \varphi(\vec{x}) p(x_1)...p(x_n)$ and $g_n[p] = \int d\vec{x}\, g(\vec{x}) p(x_1)...p(x_n)$. By convention we set $g_0[p] = \varphi_0[p] \equiv 1$. Let us now define two generating functions, the R-transform (actually $1/w$ plus the R-transform) and the Cauchy or Stieltjes transform,

$$\tilde{R}_{[p]}(w) = \sum_{n \geq 0} w^{n-1} g_n[p], \quad G_{[p]}(w) = \sum_{n \geq 0} w^{-n-1} \varphi_n[p].$$

By results from free probability theory, these two functions are inverses of each other, $\tilde{R}_{[p]}(G_{[p]}(w)) = w$. Since the generating function $F_0$ introduced in the text is such that $F_0[vp] = \sum_{n \geq 1} \frac{v^n}{n} g_n[p]$, we have $1 + v\partial_v F_0[vp] = v\tilde{R}_{[p]}(v)$. Let us now set $\mathbb{I}_{[p]}(y) := \int_0^1 dx\, p(x)\mathbb{I}_x(y)$. We have $\mathbb{I}_{[p]}(y) = \int_y^1 dx\, p(x)$, and the Cauchy transform can be written as

$$G_{[p]}(w) = \int_0^1 \frac{dx}{w - \mathbb{I}_{[p]}(x)}. \qquad \text{(B.3)}$$

Define now a new variable $w = w[v, p]$, such that $v = G_{[p]}(w)$. Then, using $\tilde{R}_{[p]}(G_{[p]}(w)) = w$, the equation $1 + v\partial_v F_0[vp] = v\tilde{R}_{[p]}(v)$ becomes

$$1 + v\partial_v F_0[vp] = vw.$$

Integrating w.r.t. $v$ yields (with the appropriate boundary condition $F_0[0] = 0$)

$$F_0[vp] = vw - 1 - \int_0^1 dx \log[v(w - \mathbb{I}_{[p]}(x))].$$

Indeed, computing the $v$-derivative of the l.h.s gives $v\partial_v F_0[vp] = vw - 1 + (\frac{\partial w}{\partial v})[v - \int_0^1 \frac{dx}{w - \mathbb{I}_{[p]}(x)}]$ which, using equation (B.3), becomes $v\partial_v F_0[vp] = vw - 1$. Setting $v = 1$ one obtains Eq. (B.1).

Let us now derive a differential equation for $b_z(x)$ for QSSEP. Using Eq. (B.1), the relation $b_z(x) = \frac{\delta F_0[a_z]}{\delta a_z(x)}$ becomes

$$b_z(x) = \int_0^x \frac{dy}{w - \mathbb{I}_{[a_z]}(y)}, \qquad \text{with} \qquad \int_0^1 \frac{dx}{w - \mathbb{I}_{[a_z]}(x)} = 1. \qquad \text{(B.4)}$$

Thus, $b_z(x)$ satisfies the boundary conditions $b_z(0) = 0$ and $b_z(1) = 1$. Furthermore, $1/b'_z(x) = w - \mathbb{I}_{[a_z]}(x)$ and $(1/b'_z(x))' = a_z(x)$. Using now $a_z(x) = \frac{h(x)}{z - h(x)b_z(x)}$, this gives, after some algebraic manipulation,

$$z b''(x) + h(x)(b'(x)^2 - b(x)b''(x)) = 0. \tag{B.5}$$

For $h(x) = \mathbb{I}_{x \in I}$, that is $h(x) = 0$ for $x \notin I$ and $h(x) = 1$ for $x \in I$, this yields the two differential equations Eq. (20) given in the main text.

## C  Derivation of the spectrum of $G_I$ for $I = [c, 1]$

First we present the derivation of the easier case where $I = [0, 1]$ which leads to Eq. (11) in the main text. In this case a solution of Eq. (20) with correct boundary conditions is $b_z(x) = z - z\left(\frac{z-1}{z}\right)^x$. Via Eq. (18) the resolvent becomes

$$R_I(z) = \int_0^1 dx \, z^{x-1}(z-1)^{-x}, \tag{C.1}$$

and has a branch cut at $z \in [0, 1]$. Via Cauchy's identity $\frac{1}{u \pm i\epsilon - \lambda} = PV(\frac{1}{u-\lambda}) \mp i\pi\delta(u - \lambda)$ on obtains the spectral density by $R_I(\lambda - i\epsilon) - R_I(\lambda + i\epsilon) = 2i\pi\ell_I d\sigma_I(\lambda)$ and we find $d\sigma_I(\lambda) = \frac{d\lambda}{\pi}\int_0^1 dx \, \sin(\pi x)\lambda^{x-1}(1-\lambda)^{-x}$. Integrating over $x$ leads to Eq. (11).

In the case of $I = [c, 1]$, a solution of Eq. (20) with correct boundary conditions is

$$b_z(x) = \begin{cases} \alpha x, & \text{if } x < c, \\ z - (z-1)Q(z)^{y-1}, & \text{if } x > c, \end{cases} \tag{C.2}$$

with $x = c + y(1-c)$. The coefficients $\alpha$ and $Q(z)$ are determined by continuity of $b_z$ and $b'_z$ at $x = c$, hence

$$zQ = (z-1) + c\alpha Q, \qquad (1-c)\alpha Q = (1-z)\log Q.$$

Eliminating $\alpha$,

$$\bar{c}\log Q = 1 + wQ, \quad \text{with} \quad \bar{c} = \frac{c}{1-c}, \quad w = \frac{z}{1-z}. \tag{C.3}$$

This specifies $Q(w)$ or equivalently $Q(z)$. The interesting part of the resolvent will thus be $\ell_I \int_0^1 \frac{dy}{z-1} Q(z)^{1-y}$. And we have to understand the analytic structure of $Q(z)$. We first look for the domain in which (C.3) has real (positive) solution. For $w$ real, this occurs for $w \in (-\infty, w_l]$ with $w_l = \bar{c}e^{-1/c}$. Graphically this is the value of $w$ where $1 + wQ$ is tangent to $\bar{c}\log Q$. Hence for $z \in (-\infty, z_l] \cup [1, +\infty)$, (C.3) has a real positive solution ($Q > 0$), with $z_l = \frac{w_l}{1+w_l} \in [0, 1]$, namely

$$z_l = \frac{c}{c + (1-c)e^{1/c}}. \tag{C.4}$$

Thus, (C.3) has no real solution on the interval $[z_l, 1]$. Let us assume that $(z_l, 1)$ is the branch cut and hence the support of the eigenvalues which is bigger than the sub-system interval $I = [c, 1] \subset (z_l, 1)$. On the branch cut, i.e. for $z \in (z_l, 1)$ or equivalently for $w \in (w_l, \infty)$, there are two complex conjugated solutions. Let $Q = e^{1/c} r e^{i\theta}$, $r > 0$ and define $w = w_l\xi$, $\xi > 1$ parametrizing the branch cut. Then real and imaginary part of Eq.(C.3) become

$$1 + \log r = r\xi\cos\theta, \quad \theta = r\xi\sin\theta. \tag{C.5}$$

We can now compute the spectral density by looking at the discontinuity of the integrand $(z-1)^{-1}Q(z)^{1-y}$. We have $Q(\lambda \pm i\epsilon) = e^{1/c} r_\lambda e^{\pm i\theta_\lambda}$, thus (after the change of variable $y \to 1-y$)

$$d\sigma_{[c,1]}(\lambda) = \mathbb{I}_{\lambda\in[z_l,1]} \int_0^1 \frac{dy}{\pi}(1-\lambda)^{-1}(e^{1/c}r_\lambda)^y \sin(y\theta_\lambda)\,d\lambda\,.$$

Integrating over $y$ yields

$$d\sigma_{[c,1]}(\lambda) = \frac{d\lambda}{\pi(1-\lambda)}\frac{\theta_\lambda + (e^{1/c}r_\lambda)\log(e^{1/c}r_\lambda)\sin\theta_\lambda - (e^{1/c}r_\lambda)\theta_\lambda\cos\theta_\lambda}{\theta_\lambda^2 + [\log(e^{1/c}r_\lambda)]^2}\mathbb{I}_{\lambda\in[z_l,1]}\,.$$

We can eliminate the trigonometric function using the relations satisfied by $r_\lambda$ and $\theta_\lambda$, to get

$$d\sigma_{[c,1]}(\lambda) = \frac{d\lambda}{\pi\lambda(1-\lambda)}\frac{\theta_\lambda}{\theta_\lambda^2 + [\log(e^{1/c}r_\lambda)]^2}\mathbb{I}_{\lambda\in[z_l,1]}\,. \tag{C.6}$$

## D Illustration of the variational principle on Wigner's matrices

Wigner's matrices are characterized by the vanishing of its associated free cumulants of order strictly bigger than two. Thus, for Wigner matrices only $g_1$ and $g_2$ are non vanishing and both are $x$-independent. All $g_k$, $k \geq 3$ are zero. Without lost of generality we can choose $g_1 = 0$ and we set $g_2 = \sigma^2$, because the matrix elements of Wigner's matrices are independent Gaussian variables with zero mean and variance $\mathbb{E}[X_{ij}X_{ji}] = N^{-1}\sigma^2$. Then $F_0[p] = \frac{\sigma^2}{2}\int dx\,dy\,p(x)p(y)$ and $R_0[p] = \sigma^2\int dx\,p(x)$. For the whole interval (considering a subset will be equivalent), the saddle point equations become (for $h = 1$)

$$a(x) = \frac{1}{z - b(x)}\,, \qquad b(x) = \sigma^2 A, \quad \text{with} \quad A = \int dx\,a(x)\,.$$

The functions $a$ and $b$ are $x$-independent. This yields a second order equation for $A$, namely $A(z - \sigma^2 A) = 1$. Solving it, with the boundary condition $A \sim \frac{1}{z} + \cdots$ at $z$ large, gives

$$A = \frac{1}{2\sigma^2}\left(z - \sqrt{z^2 - 4\sigma^2}\right)\,.$$

Thus the cut is on the interval $[-2\sigma, +2\sigma]$ and the spectral density is

$$d\sigma(\lambda) = \frac{d\lambda}{2\pi\sigma^2}\sqrt{4\sigma^2 - \lambda^2}\,\mathbb{I}_{\lambda\in[-2\sigma,+2\sigma]}\,.$$

Of course, that's Wigner's semi-circle law. Similar method can be applied to standard sub-blocks of random matrix ensembles such as Haar, Wishart, etc. ensembles (see [26] for more details).

## E Numerical method for simulating QSSEP

Sine QSSEP is a quadratic model for each realization of the noise (see Eq. 2) everything can be expressed in terms of the two point function $G_{ij}(t) = \text{Tr}(\rho_t c_i^\dagger c_j)$. This matrix evolves according to

$$G(t + dt) = e^{i\,dh_t}G(t)e^{-i\,dh_t} + \mathcal{L}(G)dt\,, \tag{E.1}$$

where

$$dh_t = \begin{pmatrix} 0 & dW_t^1 & & \\ d\overline{W}_t^1 & \ddots & & \ddots \\ & \ddots & & \ddots & dW_t^{N-1} \\ & & d\overline{W}_t^{N-1} & & 0 \end{pmatrix}\,, \tag{E.2}$$

and

$$[\mathcal{L}(G)]_{ij} = \sum_{p \in 1,N} \left( \delta_{pi}\delta_{pj}\alpha_p - \frac{1}{2}(\delta_{ip} + \delta_{jp})(\alpha_p + \beta_p)G_{ij} \right), \qquad \text{(E.3)}$$

represents the effect of the boundary reservoirs at densities $n_a = \frac{\alpha_1}{\alpha_1 + \beta_1}$ and $n_b = \frac{\alpha_N}{\alpha_N + \beta_N}$. The complex Brownian increments $dW_t^j$ are approximated by choosing a finite time step $dt$ with $t_k = k\,dt$. Then, $dW_{t_k}^j \approx x_{k,j} + iy_{k,j}$ where each $x_{k,j}, y_{k,j} \sim \mathcal{N}(0, dt/2)$ is an independent real Gaussian variable with mean zero and variance $dt/2$. Iterating Eq. (E.1) one obtains the evolution of $G$ up to any time. Instead of repeating this for different realizations of the noise, we time average the evolution of $G$ once its distribution has reached a steady state. This is possible because the QSSEP trajectories in the steady state are ergodic: Ensemble averages can be replaced by time averages.

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
