# Peer review of "Exact Entanglement in the Driven Quantum Symmetric Simple Exclusion Process"

_SciPost Physics, doi:SciPost Phys. 15, 175 (2023)_

## Round 4 · Referee Report · Anonymous · 2023-9-8

Report
Some more comments to suggest improvements to the clarity of this paper:
It appears that the results are for the NESS (nonequilibrium steady state) and not for any transient states, since no dependence on initial state or on time is mentioned. This should be made more explicit. Perhaps a few lines above Eq. 3 it could say "all information about the NESS" (instead of system), and there also say that the NESS is unique (thus independent of the initial state) and is a Gaussian state.
In the short 3rd paragraph of the Intro it says "...mutual information between two subregions of a thermal state....". I think the authors have in mind some specific geometry, perhaps of two contiguous subregions that share a boundary, since if the subregions are distant the mutual information in a nonzero temperature Gibbs state falls off exponentially with the distance between the two subregions. They probably mean the two subregions have a shared boundary and the mutual information is proportional to the area of that boundary. The authors also need to be more precise here about what types of thermal states they are talking about. Later in the paper they consider eigenstates obeying the ETH, which are thus certainly thermal states, and these do have volume law mutual information if the two subregions are large enough fractions of the full system. So here I think the authors mean certain specific types of thermal mixed states, perhaps Gibbs/Boltzmann states of some type, or microcanonical mixed states of high enough entropy. This again should be made more precise and explicit.

---

## Round 4 · Referee Report · Anonymous · 2023-9-13

Report
The authors in the new submission have addressed the remarks I made in my report. However, I still find the modification at the end of page 4 regarding the special case of a system in equilibrium to be poorly phrased. In the paper, if λ is distributed according to the special case, then n_a + (n_b - n_a) λ should be distributed according to the general case. This means that for n_a = n_b, the distribution should be a Dirac centered on n_a, and not a constant uniform distribution as mentioned in the paper. It would be helpful to phrase this part correctly.
Lastly, I noticed a couple of typos in the added annex: "W_t" should be corrected to "dw_t" in the matrix, and "sine" should be corrected to "since."

---

## Round 4 · Author Response

List of changes
Changes made in response to the 2nd referee's comments (as suggested by the editor):
1. "In the introduction, it is suggested that the volume law of mutual entropy is an out-of-equilibrium phenomenon since systems at equilibrium follow an area law at zero temperature, according to ref [7]. It would be helpful to confirm this idea on QSSEP, by singling out the limit of equal reservoir densities, one expects a breakdown of the area law in this limit since the model becomes at equilibrium. It would be constructive to provide insight into how this can occur within the formalism presented in the paper and to refer to relevant existing literature if this limit has already been addressed."
-> We explain the equilibrium case in the 3rd sentence after Eq. (7)
2. "It's not clear in the paper why specifically the Reyni entropy of order 2 has been chosen."
-> We comment on this in the 3rd sentence after Eq. (6)
3. "The methodology employed for the numerical simulation is completely absent."
-> We added an appendix E explaining the numerical methodology for simulating QSSEP
4. "In section 2, the usage of indices belonging to a continuous real interval for a discrete model is a bit confusing and becomes only clear in section 3, Clarifying the relationship between these indices should be considered upon their introduction."
-> This is now clarified in the last sentence on p. 3
5. "In section 2, the conclusion of the mutual information scaling as the volume of the subregion seems to rely on the figure, One can argue that there might be other higher non-linear terms with small coefficients. The precise meaning of the scaling in terms of the size of the subsystem and the total system is not stated clearly enough."
-> The volume scaling does not rely on the figure but only on analytics and we explain why this is the case in a new paragraph above Eq. (7) starting with "However,..."
6. "There is a missing minus sign in the definition of entanglement entropy in the introduction."
-> This is now corrected
7. "In the introduction, it might be better to refer to the area of the boundaries of the subregion, rather than the area of the subregion itself, as it can be confusing for readers who are not in the field."
-> We now refer to the "area of the boundary of a subregion"
Other changes made:
- >Added an explicit formula for the van Neumann entropy in Eq. (5)
- >Added "in its steady state (reached after long times)" in the 1st sentence on page 3

---

## Round 4 · List of Changes

Changes made in response to the 2nd referee's comments (as suggested by the editor):
1. "In the introduction, it is suggested that the volume law of mutual entropy is an out-of-equilibrium phenomenon since systems at equilibrium follow an area law at zero temperature, according to ref [7]. It would be helpful to confirm this idea on QSSEP, by singling out the limit of equal reservoir densities, one expects a breakdown of the area law in this limit since the model becomes at equilibrium. It would be constructive to provide insight into how this can occur within the formalism presented in the paper and to refer to relevant existing literature if this limit has already been addressed."
-> We explain the equilibrium case in the 3rd sentence after Eq. (7)
2. "It's not clear in the paper why specifically the Reyni entropy of order 2 has been chosen."
-> We comment on this in the 3rd sentence after Eq. (6)
3. "The methodology employed for the numerical simulation is completely absent."
-> We added an appendix E explaining the numerical methodology for simulating QSSEP
4. "In section 2, the usage of indices belonging to a continuous real interval for a discrete model is a bit confusing and becomes only clear in section 3, Clarifying the relationship between these indices should be considered upon their introduction."
-> This is now clarified in the last sentence on p. 3
5. "In section 2, the conclusion of the mutual information scaling as the volume of the subregion seems to rely on the figure, One can argue that there might be other higher non-linear terms with small coefficients. The precise meaning of the scaling in terms of the size of the subsystem and the total system is not stated clearly enough."
-> The volume scaling does not rely on the figure but only on analytics and we explain why this is the case in a new paragraph above Eq. (7) starting with "However,..."
6. "There is a missing minus sign in the definition of entanglement entropy in the introduction."
-> This is now corrected
7. "In the introduction, it might be better to refer to the area of the boundaries of the subregion, rather than the area of the subregion itself, as it can be confusing for readers who are not in the field."
-> We now refer to the "area of the boundary of a subregion"
Other changes made:
- >Added an explicit formula for the van Neumann entropy in Eq. (5)
- >Added "in its steady state (reached after long times)" in the 1st sentence on page 3

---

## Round 5 · Author Response

Thanks for the minor comments from the referees. We implemented them all, see below.

---

## Round 5 · List of Changes

• abstract, 3rd sentence: added "in the steady state"
  • Introduction, 3rd paragraph, 2nd sentence: added "adjacent" and replaced "thermal"->"Gibbs"
  • Last sentence on p. 3: added "In this article we are only interested in the steady state distribution of coherences, which is unique regardless of the initial condition."
  • Last sentence on p. 4: modified "This also shows that in the equilibrium case where na = nb, all eigenvalues are equal to na. That is, dσc1,c2 = δ(λ−na)dλ is independent of the interval [c1, c2]"

---

## Editorial Decision

published